# Efficient and Accurate Spatial-Temporal Denoising Network for Low-dose CT Scans

**Leihao Wei** [1,2]                                     LEIHAOWEI@ENGINEERING.UCLA.EDU

**William Hsu** [2]                                           WHSU@MEDNET.UCLA.EDU

[1] *Electrical and Computer Engineering, University of California, Los Angeles, USA*

[2] *Medical & Imaging Informatics, Department of Radiological Sciences, David Geffen School of Medicine at UCLA, USA*

## Abstract

While deep-learning-based imaging denoising techniques can improve the quality of low-dose computed tomography (CT) scans, repetitive 3D convolution operations cost significant computation resources and time. We present an efficient and accurate spatial-temporal convolution method to accelerate an existing denoising network based on the SRResNet. We trained and evaluated our model on our dataset containing 184 low-dose chest CT scans. We compared the performance of the proposed spatial-temporal convolution network to the SRResNet with full 3D convolutional layers. Using 8-bit quantization, we demonstrated a 7-fold speed-up during inference. Using lung nodule characterization as a driving task, we analyzed the impact on image quality and radiomic features. Our results show that our method achieves better perceptual quality, and the outputs are consistent with the SRResNet baseline outputs for some radiomics features (31 out of 57 total features). These observations together demonstrate that the proposed spatial-temporal method can be potentially useful for clinical applications where the computational resource is limited.

**Keywords:** image restoration, efficiency, network quantization, image quality, radiomics

## 1. Introduction

Computed tomography (CT) scans provide a detailed characterization of chest anatomy for radiologists to identify lesions in the lung. However, in practice, CT acquisitions are not standardized. Given that higher radiation exposure comes with the risk of harmful radiation, the trend has been to acquire lower dose images at the cost of noisier images. Recent developments in deep learning-based image denoising have yielded a number of approaches to recover high-resolution details from lower resolution inputs. Prior studies have also demonstrated that 3D convolutions compared to 2D convolutions achieve better image quality (Shan et al., 2018). However, one barrier is that such a method is computationally expensive. In this paper, we utilize the spatial and temporal correlation in CT scans to introduce an efficient neural network architecture, Spatial-Temporal ResNet (STResNet) that restores the high-resolution details from low-dose CT images. Our goal is to achieve the same level of accuracy as the standard 3D SRResNet while improving its efficiency.

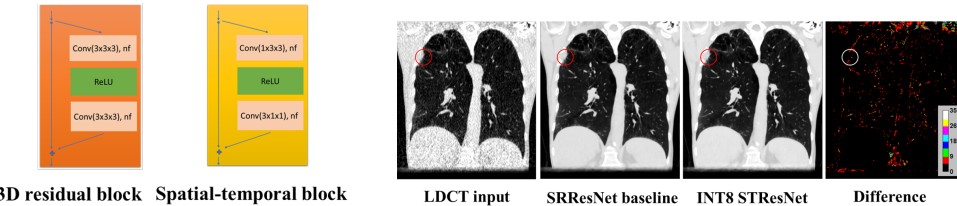

**3D residual block**  **Spatial-temporal block**     **LDCT input**  **SRResNet baseline**  **INT8 STResNet**  **Difference**

Figure 1: Convolutional blocks and results. A nodule ROI is highlighted in the circle.

## 2. Method and Data

Inspired by Enhanced Deep Residual Networks (EDSR) (Lim et al., 2017), we implemented a baseline denoising network based on SRResNet using fully 3D convolutional layers with a series of residual in residual blocks with convolutional and activation layers. Since CT scans are 3D volumes consisting of multiple slices, each slice can be treated as a frame at a time step. For each pixel in a slice, spatial and temporal correlation exists in adjacent frames along the temporal dimension. Hence, in STResNet, we decompose a full 3D convolution with $3 \times 3 \times 3$ kernel into two smaller convolutions, each with a spatial and temporal kernel. As illustrated in Figure 1, 3D convolutional blocks are replaced with spatial ($1 \times 3 \times 3$) and temporal ($3 \times 1 \times 1$) convolutional blocks (Li et al., 2019).

We demonstrate the differences in efficiency and accuracy using a dataset of low-dose CTs acquired for lung cancer screening, acquired at an equivalent dose about 2mGy. The standard condition was acquired at 100% dose and reconstructed using a medium kernel and 1.0 mm slice thickness, which reflects the parameters that are currently recommended for lung cancer screening. In the test set of 84 patients, 42 scans (50%) were found to have a total of 68 lung nodules. Lower-dose CT images were reconstructed from raw data of standard acquisitions using a physics-based model that simulates noise characteristics as well as reconstruction artifacts that are equivalent to 10% of the standard dose and at 2.0mm slice thickness. Data were split into 80/20/84 for training/validation/test. We adapted the NVIDIA APEX mixed-precision training package to further improve the training speed with mixed precision on GPU. We also introduced 8-bit low-precision quantization (Jacob et al., 2018) to SRResNet and STResNet to achieve faster inference on CPU.

## 3. Evaluation and Results

Our method was validated using image quality metrics such as peak signal-to-noise ratio (PSNR), structural similarity (SSIM) and Learned Perceptual Image Patch Similarity (LPIPS) (Zhang et al., 2018). In Table 1, our STResNet achieved better PSNR and SSIM compared to SRResNet in full precision (FP32) inference. Quantization (INT8) was shown to negatively impact image quality using PSNR and SSIM as metrics (a decrease of 0.16dB and 0.0152 respectively). However, compared to the baseline model, STResNet with 8-bit quantization achieved better perceptual quality (0.3555 vs. 0.3653). As shown in Figure 1, the difference between the result of baseline and 8-bit quantized STResNet at nodule ROI is visually imperceptible. During inference tasks on CPU, our quantized STResNet achieves up to 7.11 times speed-up compared to the standard SRResNet. Using STResNet alone

achieves a speed up by a factor of 1.67. A similar trend is observed during training on GPU with up to 2 times speed up when using STResNet FP16 versus SRResNet FP32.

To assess differences in radiomic features, we selected 57 first-order intensity, gray-level concurrence matrix (GLCM), gray-level run length matrix (GLRLM), and gray-level size zone matrix (GLSZM) features to study the impact of feature values on nodules by using different combinations of networks and precision. In Figure 2, we found 54% of feature values from the outputs of quantized STResNet were still consistent to the baseline distribution.

Table 1: Image quality metrics and speed-up factors to baseline. * CPU results

| | | ↑ PSNR(dB) | ↑ SSIM | ↓LPIPS | Inference time (sec) | Training time per iter (sec) | Inference Speed-up | Training Speed-up |
|---|---|---|---|---|---|---|---|---|
| FP32 | SRResNet (baseline) | 31.31±0.30 | 0.7216±0.0113 | 0.3635±0.0074 | 27.4(446.7*) | 6.5 | N/A | N/A |
| | **STResNet** | 31.91±0.44 | 0.7265±0.0110 | 0.3715±0.0075 | 14.4(267.0*) | 3.9 | 1.67 | 1.65 |
| FP16 | SRResNet | 32.39±0.52 | 0.7277±0.0111 | 0.3640±0.0075 | 13.8 | 4.9 | N/A | 1.31 |
| | **STResNet** | 32.60±0.64 | 0.7259±0.0111 | 0.3732±0.0076 | 17.0 | 3.2 | N/A | **2.04** |
| INT8 | SRResNet | 31.15±0.28 | 0.7064±0.0109 | 0.3501±0.0075 | 108.7* | N/A | 4.11 | N/A |
| | **STResNet** | 31.11±0.30 | 0.7135±0.0109 | 0.3555±0.0076 | 62.8* | N/A | **7.11** | N/A |

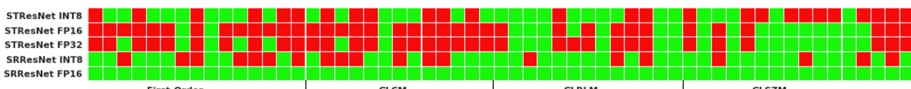

Figure 2: Radiomic features test. Red/Green indicates significant/non-significant difference to baseline via paired t-test with p < 0.05.

## 4. Discussion

We trained and evaluated our efficient and accurate network architecture called STResNet for low-dose CT denoising. Through our study, we demonstrated that STResNet reduces the training and inference time compared to SRResNet. We also showed that 8-bit quantization produced outputs that had minimal perceptual differences despite the information loss of computing a 12-bit CT scan using 8-bit quantized network weights. We note in our results that some radiomic features have statistically significant differences in distribution compared to feature values calculated from SRResNet outputs. Further study is required to assess the impact of 8-bit quantization and STResNet assumptions on downstream tasks such as machine learning algorithm performance. As part of future work, we will investigate the impact of using the efficient network architecture on clinical-driven tasks such as lung nodule detection or diffuse lung disease quantification.

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
