# OpenReview forum: "Efficient and Accurate Spatial-Temporal Denoising Network for Low-dose CT Scans"
_MIDL.io/2021/Conference/Short — MIDL 2021 Poster_

### Official Review · Reviewer_8FCc · 2021-04-29

**Confidence:** 4
**Final Rating:** 3

**Summary:**

The paper proposes a new convolutional neural network for denoising of 3D computed tomography volumes that decreases the inference time by a factor of 7 by 1) splitting up the computational expensive 3D convolution ($3\times 3\times 3$ kernel) into a spatial ($1\times 3\times 3$ kernel) and a temporal ($3\times 1\times 1$ kernel) and 2) using INT8 precision for the network's weights. The results of the so-called STResNet (Spatial Temporal ResNet) are compared to the standard one concerning PSNR, SSIM and LPIPS metric, where mixed results are reported. Further, radiomic features (lung nodules) are analysed in all network's outputs, where only 54% of regions show a similar texture as the FP32 ResNet's output.

**Strengths:**

The lack of computational ressources is a challenging problem in medical imaging that is properly addressed in this paper. The idea of splitting the computational expensive 3D convolution into less time consuming 2D spatial and temporal convolutions decreases the inference time significantly. While almost cutting inference and training time in half with both networks in FP32 precision mode the STResNet achieves higher PSNR, SSIM and LPIPS scores compared to the standard SRResNet.

**Weaknesses:**

The quantification of texture comparison in lung nodule ROIs could be more clearly explained. Further, the speed-ups are interchangebly given on CPU and GPU instead of a stringent notion. The lung nodule ROIs are characterized as "visually imperceptible" the texture analysis shows different, with only 54% of ROIs showing no significant deviation.

**Deanonymize Review:**

no

**Detailed Comments:**

- It should be gray level co-occurence matrix instead of gray level concurrence matrix.
- Could the proposed method also be used for other modalities?
- Will the code for the radiomic features test also be released?
- Were the results examined by a doctor? Has the difference in texture values concerning GLCM, GLRLM, and GLDZM a meaningful influence on doctor's diagnoses?
- Is there no inference speedup for FP16 precision and is INT8 training not possible?
- LPIPS uses models that were trained on natural images. Its application in medical imaging seems limited therefore.

**Justification Of The Rating:**

The paper is a valuable contribution to the field, reducing memory requirements and by that decreasing the computation time of models can lift artificial intelligence into real world time critical applications.
The paper is well written and clearly explains the proposed solution.
Revisions, as described above, can further improve the paper.

**Paper Type:**

both

**Special Issue:**

no

---

### Official Review · Reviewer_X7bL · 2021-05-01

**Confidence:** 4
**Final Rating:** 3

**Summary:**

The paper proposes a spatial-temporal convolution module to reduce the computational burden of traditional denoising networks for low-dose CT scans.  The authors claim that the contribution of the method is that it can be potentially useful for clinical applications where the computational resource is limited. Also, the experiments verify the effectiveness of the proposed methods.

**Strengths:**

1. The results showed the effectiveness of the method with encouraging speed-up.
2. Using radiomic features to verify the image consistency is also interesting.
3. Comparing the proposed methods in different FP precision really makes the results more convincing.

**Weaknesses:**

1. The motivation here is not clear, as usually, the post-processing time for certain medical imaging tasks is not that sensitive. That is to say, it is meaningless to get a speed up by sacrificing the accuracy.
2. When comes to the network with different FP precision, I believe there is a ton of work published. Thus, I would suggest the author has a more comprehensive literature review before they turn this short paper to long paper.

**Deanonymize Review:**

no

**Detailed Comments:**

The authors are encouraged to make the motivation clearer. At least from the application of CT-scan used here, it is not necessary to reduce the network inference time with a potential impact on the accuracy.


**Justification Of The Rating:**

This paper is well-written and easy to follow. It focused on an important task of improving the quality of the low-dose CT image. Also, the authors used the proposed method to improve the inference speed. However, the motivation for designing such a network/improve the inference speed is not clear.


**Paper Type:**

validation/application paper

**Special Issue:**

no

---

### Meta-Review · Area_Chair_bPVP · 2021-05-09

**Recommendation:** Accept (Poster)
**Confidence:** 5

**Metareview:**

Nice work addressing practical problems that received positive remarks from both reviewers and authors have released source code. I recommend acceptance.

---

### Decision · Program_Chairs · 2021-05-11

Accept (Poster)